

# Global sensitivity analysis of a dynamic model for gene expression in *Drosophila* embryos

Gregory D. McCarthy[1], Robert A. Drewell[2] and Jacqueline M. Dresch[3,4]

[1] School of Natural Science, Hampshire College, Amherst, MA, USA
[2] Biology Department, Clark University, Worcester, MA, USA
[3] Department of Mathematics, Amherst College, Amherst, MA, USA
[4] Department of Mathematics and Computer Science, Clark University, Worcester, MA, USA

## ABSTRACT

It is well known that gene regulation is a tightly controlled process in early organismal development. However, the roles of key processes involved in this regulation, such as transcription and translation, are less well understood, and mathematical modeling approaches in this field are still in their infancy. In recent studies, biologists have taken precise measurements of protein and mRNA abundance to determine the relative contributions of key factors involved in regulating protein levels in mammalian cells. We now approach this question from a mathematical modeling perspective. In this study, we use a simple dynamic mathematical model that incorporates terms representing transcription, translation, mRNA and protein decay, and diffusion in an early *Drosophila* embryo. We perform global sensitivity analyses on this model using various different initial conditions and spatial and temporal outputs. Our results indicate that transcription and translation are often the key parameters to determine protein abundance. This observation is in close agreement with the experimental results from mammalian cells for various initial conditions at particular time points, suggesting that a simple dynamic model can capture the qualitative behavior of a gene. Additionally, we find that parameter sensitivites are temporally dynamic, illustrating the importance of conducting a thorough global sensitivity analysis across multiple time points when analyzing mathematical models of gene regulation.

# INTRODUCTION

## Gene regulation in embryonic development

Embryonic development in animals is very precisely controlled by a network of regulatory proteins (*Davidson, 2010*; *Peter and Davidson, 2011*). For any particular protein, the exact level of expression at a specific time point can be crucial to the proper development of the organism (*Davidson & Levine, 2008*). Within each cell of the developing embryo, protein abundance is a function of two key molecular events: transcription and translation. Transcription is the process of reading a gene in a DNA template to produce a messenger RNA (mRNA), while translation is the process of reading the mRNA to produce a protein

Corresponding author
Jacqueline M. Dresch,
jdresch@clarku.edu

product. A simple, but long-standing question in Biology is the following: Which is contributing more to the variance in protein levels in cells, transcription or translation?

## Work analyzing the importance of transcription in mammalian cells

Many experimental studies have been conducted in an attempt to understand the roles of transcription and translation in regulating the dynamic nature of mRNA and protein concentrations (*Maier, Guell & Serrano, 2009*; *Vogel et al., 2010*; *Schwanhausser et al., 2011*; *Beck et al., 2011*; *Bantscheff et al., 2012*; *Vogel & Marcotte, 2012*; *Li, Bickel & Biggin, 2014*). One such recent detailed study, conducted by the Biggin lab using data from ubiquitously expressed (housekeeping) genes in cultured mammalian cells, aimed to improve existing quantification of protein abundances through statistical analysis of the impact of experimental error (*Li, Bickel & Biggin, 2014*). In this study, a two-part regression procedure was used to derive new estimates of protein abundance from the 2011 data set of *Schwanhausser et al. (2011)*. Using these new, corrected measurements of protein abundances, along with previous measurements of mRNA and protein degradation rates, they were able to determine the relative importance of transcription, translation, mRNA degradation, and protein degradation (see Fig. 1A). This analysis is referred to as the measured protein error strategy. The result of this procedure found that for the 4,212 genes considered, transcription contributed the most to protein abundance ($\sim$38%), with translation contributing slightly less ($\sim$30%), followed by mRNA degradation ($\sim$18%) and protein degradation ($\sim$14%) (*Li, Bickel & Biggin, 2014*). This result is important to note, as it differs drastically from Schwanhausser et al.'s original conclusion that translation accounts for the largest contribution ($\sim$55%) to overall variance in the cellular abundance of proteins (*Schwanhausser et al., 2011*; *Li, Bickel & Biggin, 2014*).

## Existing mathematical models of gene expression

Due to the quantitative nature of gene regulation in the embryo, and the advent of new experimental techniques giving rise to massive amounts of mRNA and protein concentration data, various mathematical models have been derived and implemented to help understand the complexity that lies within developmental gene regulatory networks. These models range from static models, considering only transcription at a single time point in development in a single cell, to dynamic spatio-temporal models that incorporate transcription, translation, diffusion, and decay rates for a network of genes that regulate one another over a continuous time frame (*Jaeger et al., 2004*; *Santillan & Mackey, 2004*; *Bintu et al., 2005*; *Janssens et al., 2006*; *Zinzen et al., 2006*; *Segal et al., 2008*; *Gertz, Siggia & Cohen, 2009*; *Fakhouri et al., 2010*; *He et al., 2010*; *Bieler, Pozzorini & Naef, 2011*; *Janssens et al., 2013*; *Ilsley et al., 2013*; *Dresch et al., 2013*; *Samee & Sinha, 2014*).

To accurately model protein abundance in a metazoan animal, such as a mouse or fruit fly, one must consider both spatial and temporal dynamics in the developmental system. One such model that we developed uses a discretized reaction–diffusion equation to model concentrations of mRNA and protein in a developing *Drosophila* embryo across *n* nuclei (*Dresch et al., 2013*). This model not only incorporates terms for mRNA and protein
**A. Reaction**

**B. Diffusion**

**Figure 1 Schematic of the biological procceses represented in the ODE model.** In (A) the Reaction terms of the model are illustrated. These include the synthesis of new mRNA through transcription, the synthesis of new protein through translation of mRNA, mRNA decay, and protein decay. In B, the Diffusion terms of the model are illustrated. These include both mRNA diffusion and protein diffusion to/from neighboring nuclei in an early *Drosophila* embryo.

synthesis and decay, but also diffusion of these molecules in the developing embryo. This is particularly important in *Drosophila* development as the early stages of embryogenesis are marked by 13 mitotic (nuclear) divisions in the absence of cellular divisions, resulting in a multinucleate syncytial embryo (see Fig. 1B).

In its simplest form, the model can be written as:

$$\frac{dy_{a,i}}{dt} = S_{a,i}(Y) + D_a(y_{a,i+1} - 2y_{a,i} + y_{a,i-1}) - \lambda_a y_{a,i} \qquad (1)$$

where $a$ represents the specific mRNA or protein that the equation corresponds to, $i$ represents the specific nucleus, $D_a$ and $\lambda_a$ are the corresponding diffusion and decay rates, and $Y$ represents the entire vector of mRNA and protein concentrations within the system being modeled.

Many similar models have been used to model the expression dynamics of the gap gene system in the developing *Drosophila* embryo (*Jaeger et al., 2004*; *Okabe-Oho et al., 2009*; *Ashyraliyev et al., 2009*; *Bieler, Pozzorini & Naef, 2011*; *Holloway et al., 2011*; *Janssens et al., 2013*; *Holloway & Spirov, 2015*). Although these models all rely on an underlying reaction–diffusion framework, they vary greatly in their implementation. Both deterministic (*Jaeger et al., 2004*; *Ashyraliyev et al., 2009*; *Bieler, Pozzorini & Naef, 2011*; *Janssens et al., 2013*) and stochastic (*Okabe-Oho et al., 2009*; *Holloway et al., 2011*; *Holloway & Spirov, 2015*) models have been able to accurately predict the effects of particular perturbations to the network. Stochastic models of *hunchback* regulation have been used to shed light on
the underlying factors that reduce noise and promote stability of the *hunchback* gradient. These include the number or arrangement of BICOID and KRUPPEL binding sites within the regulatory sequences that control transcription of the *hunchback* gene as well as protein diffusion (*Okabe-Oho et al., 2009*; *Holloway et al., 2011*; *Holloway & Spirov, 2015*). In this study, we focus on the broad impacts of transcription, translation, diffusion, and decay, and do not consider specific transcription factor binding sites within regulatory DNA sequences. Thus, we focus on a deterministic model and do not include any stochasticity.

## Global parameter sensitivity analysis and HDMR

To help develop a better understanding of the model parameters, including how their values impact the model output (protein abundance) and how one might interpret that impact with respect to the biological system, parameter sensitivity analysis is needed. Parameter sensitivity analysis refers to a mathematical analysis of the change in model output as a result of variation in the input parameter values (*Frey & Patil, 2002*; *van Riel, 2006*; *Tang et al., 2006*; *Dresch et al., 2010*; *Ay & Arnosti, 2011*; *Jarrett et al., 2014*). This analysis can be done locally, at a particular point in parameter space, or globally across the entire parameter space.

Local parameter sensitivity analyses are typically implemented by simply computing or approximating the partial derivative of the objective function at a particular point in parameter space to determine how the function changes locally with respect to small variations of a particular parameter (*van Riel, 2006*; *Dresch et al., 2010*; *Reeves & Fraser, 2009*). The major advantages to adopting a local approach are that it is straightforward, often easy to interpret, and computationally inexpensive. However, a significant limitation of local methods is that when dealing with a large parameter space focusing on a single point in that space may not be representative of much of the overall parameter space. In contrast, global methods allow one to calculate parameter sensitivities while considering the full range of parameter space (*Jarrett et al., 2014*). Most global methods also have the ability to calculate higher order sensitivities, which capture interactions between parameters. This can be challenging to do with a local method, especially when parameter space is large and many different parameter combinations within that space lead to valid model outputs. Therefore, in this study we focus on a global method for parameter sensitivity analysis for our model of gene expression in the *Drosophila* embryo.

Higher Dimensional Model Representation (HDMR) is a robust global method for calculating parameter sensitivities (*Ziehn & Tomlin, 2009*; *Dresch et al., 2010*). This method uses the Monte Carlo integration method to decompose the model output, $f(x)$, often referred to as the objective function, into terms of increasing dimensionality with respect to the parameters $x_1, \ldots, x_N$:

$$f(x) = f_0 + \sum_{i=1}^{N} f_i(x_i) + \sum_{1 \leq i < j \leq N} f_{ij}(x_i, x_j) + \cdots + f_{12\ldots N}(x_1, x_2, \ldots, x_N). \qquad (2)$$

In the above equation, $f_0$ is the main effect, and is approximated by the overall mean of model output over all parameter sets sampled. Each function $f_i(x_i)$ is a first-order term

representing the effect of the parameter $x_i$ acting independently on the model output. Each function $f_{ij}(x_i, x_j)$ is a second-order term representing the effect of the parameters $x_i$ and $x_j$ on the model output. These terms represent the impact pairwise parameter interactions have on determining the model output.

This approximation is done on a bounded subset of $\mathbb{R}^N$, where $N$ is the number of model parameters. The bounded subset represents the parameter space and in each dimension, corresponds to a realistic range for the given parameter. In some applications, the parameter space is known or experimentally determined using empirically determined measurements (*Ziehn & Tomlin, 2009*; *Tang et al., 2006*). In other cases, parameter space is chosen based on model assumptions and simulations (*Frey & Patil, 2002*; *Gutenkunst et al., 2007*; *Dresch et al., 2010*; *Ay & Arnosti, 2011*; *Jarrett et al., 2014*).

One of the main assumptions when using HDMR is that the objective function values are normally distributed (*Ziehn & Tomlin, 2009*). Thus, using the bounded parameter space, one generates a pseudo-random sampling using a Sobol Set (*Sobol, 1976*). The bounded parameter space must be a hypercube in $N$-dimensional space, so it can be normalized to the unit hypercube for sampling. Each set of parameter values is then used as input for a model simulation and the corresponding model output is obtained. Once this sampling has been done for all parameter sets sampled, HDMR approximates the mean of the output values, $f_0$, as well as higher order terms using orthonormal polynomial approximations. First- and second-order terms are then normalized by the total variance to obtain the main effect of each parameter and the effects of pair-wise parameter interactions, referred to as first- and second-order sensitivity indices respectively. Although this method has the ability to calculate higher-order sensitivities, it has been shown that using only first- and second-order terms is sufficient to approximate the total sensitivity in the system (*Li, Rosenthal & Rabitz, 2001*; *Liang & Guo, 2003*; *Dresch et al., 2010*).

In this study, we utilize a global HDMR analysis to investigate the sensitivity of our *Drosophila* embryo gene expression model to the individual transcription, translation, diffusion, and decay rate parameters and higher-order interactions between these parameters. In addition, we compare our results to those in mammalian cells and other studies that have attempted to model different gene expression systems (*Li, Bickel & Biggin, 2014*).

## METHODS

### Simplified model and parameters

In this study, we use a simplified version of our earlier model (*Dresch et al., 2013*), which was used to predict both mRNA and protein concentrations along a one-dimensional strip of nuclei in a developing *Drosophila* embryo.

The simplifying assumption applied in the current study is that the gene of interest has spatially uniform transcriptional activity. Thus, the transcription rate is held constant. This allows us to utilize simple numerical solvers such as Euler's method or Runga-Kutta methods, and to measure the relative importance of transcription to the model output using a single parameter, $\sigma$. Note that diffusion is discretized with respect to space and zero flux boundary conditions are used (*Dresch et al., 2013*). For $2 \leq i \leq n - 1$, where

**Table 1** This table describes all model parameters and the ranges used during the sensitivity analysis.

| Parameter | Definition | Range |
|---|---|---|
| $\sigma$ | Transcription rate | $[0.012, 2.0]$ |
| $d$ | mRNA diffusion rate | $[0.0, 1.5]$ |
| $\lambda$ | mRNA decay rate | $[0.0, 0.8]$ |
| $\tau$ | Translation rate | $[0.125, 1.0]$ |
| $\delta$ | Protein diffusion rate | $[0.0, 1.5]$ |
| $\gamma$ | Protein decay rate | $[0.4, 0.9]$ |

**Table 2** This table contains the first-order sensitivities at the middle nucleus at $t = 4$ min for a ubiquitous gene with initial concentrations of 1.0.

| | First-order sensitivity | % Contribution to variance (*Li, Bickel & Biggin, 2014*) |
|---|---|---|
| 1. Transcription | 0.32 | 0.38 |
| 2. mRNA Diffusion | 0.00 | N/A |
| 3. mRNA Decay | 0.17 | 0.18 |
| 4. Translation | 0.32 | 0.30 |
| 5. Protein Diffusion | 0.00 | N/A |
| 6. Protein Decay | 0.05 | 0.14 |

$n$ corresponds to the number of nuclei being modeled, the model equations are thus as follows:

$$\frac{dy_1}{dt} = \sigma + d(y_2 - y_1) - \lambda y_1$$

$$\frac{dy_i}{dt} = \sigma + d((y_{i+1} - y_i) + (y_{i-1} - y_i)) - \lambda y_i$$

$$\frac{dy_n}{dt} = \sigma + d(y_{n-1} - y_n) - \lambda y_n$$

$$\frac{dy_{n+1}}{dt} = \tau y_1 + \delta(y_{n+2} - y_{n+1}) - \gamma y_{n+1}$$

$$\frac{dy_{n+i}}{dt} = \tau y_i + \delta((y_{n+i+1} - y_{n+i}) + (y_{n+i-1} - y_{n+i})) - \gamma y_{n+i}$$

$$\frac{dy_{2n}}{dt} = \tau y_n + \delta(y_{2n-1} - y_{2n}) - \gamma y_{2n}.$$

Here, $y_j$ represents mRNA concentrations for $1 \leq j \leq n$ and protein concentrations for $n + 1 \leq j \leq 2n$. A schematic of the biological processes incorporated in the model is shown in Fig. 1. The model parameters are defined in Table 1. For all of the analysis and results shown in Table 2 and Figs. 2–4, 52 nuclear positions are used at approximately 50% egg height (ventral-dorsal) from approximately 10% to 90% egg length (anterior-posterior) and the numerical solver Runga-Kutta 4 is used to approximate the solutions to the system of Ordinary Differential Equations shown above.

McCarthy et al. (2015), *PeerJ*, DOI 10.7717/peerj.1022
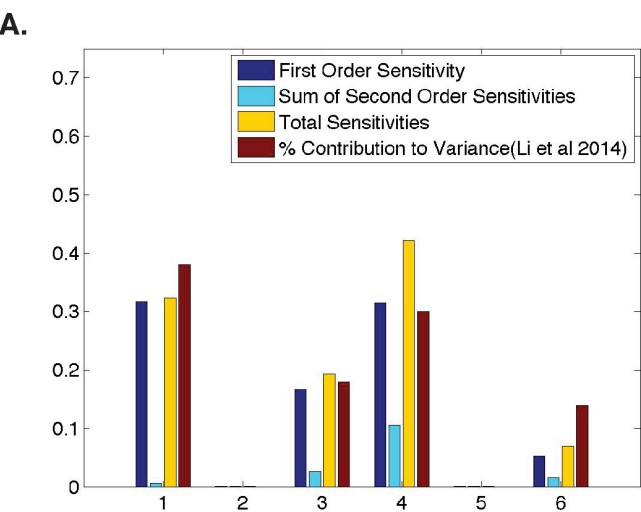

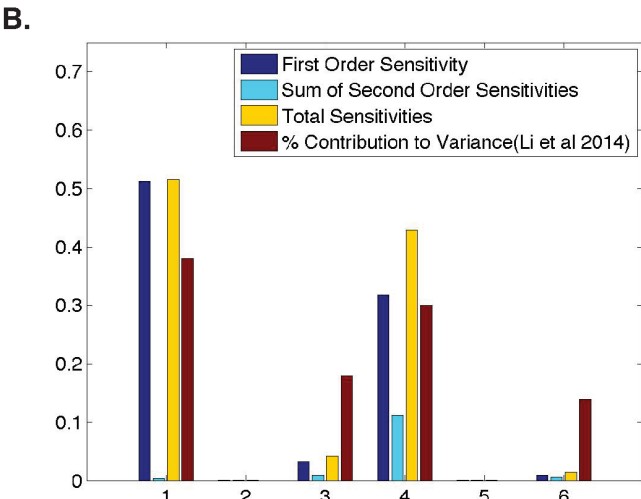

**Figure 2 Qualitative similarities between parameter sensitivities and experimental measurements.** (A) Ubiquitous gene with initial concentrations of 1.0; First and second-order sensitivities at the middle nucleus at $t = 4$ min. (B) Anterior maternally deposited gene; First and second-order sensitivities at the middle nucleus at $t = 2$ min. In both, along the $x$-axis are the parameters corresponding to: 1. Transcription, 2. mRNA Diffusion, 3. mRNA Decay, 4. Translation, 5. Protein Diffusion, and 6. Protein Decay.

## Initial conditions

Five different initial conditions are used in this study. Each initial condition corresponds to a different group of spatially expressed genes present in an early *Drosophila* embryo. The first three initial conditions all correspond to genes that are ubiquitously expressed at spatially uniform levels, such as *Zelda* in the *Drosophila* embryo. The only difference between these three initial conditions are the levels of the initial concentrations of mRNA and protein. The initial protein and mRNA concentrations used are 0, $\frac{1}{2}$, and 1. These initial
**A.**

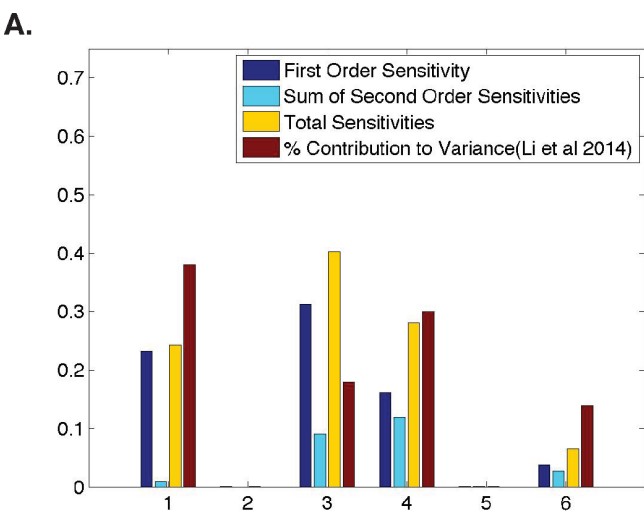

**B.**

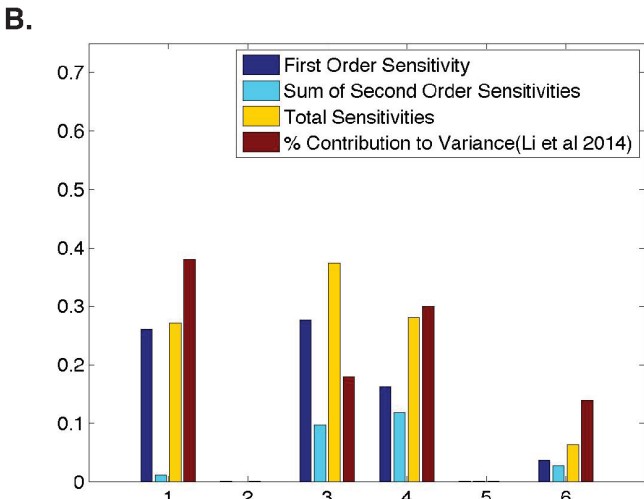

**Figure 3 Comparison of parameter sensitivities to experimental measurements at a later time point.** (A) Ubiquitous gene with initial concentrations of 1.0; First and second-order sensitivities at the middle nucleus at $t = 10$ min. (B) Anterior maternally deposited gene; First and second-order sensitivities at the middle nucleus at $t = 10$ min. In both, along the $x$-axis are the parameters corresponding to: 1. Transcription, 2. mRNA Diffusion, 3. mRNA Decay, 4. Translation, 5. Protein Diffusion, and 6. Protein Decay.

conditions allow us to compare our calculated parameter sensitivities to the contribution to variance in mammalian housekeeping genes measured by *Li, Bickel & Biggin (2014)*.

The other two initial conditions used in the study are representative of genes that are known to be extremely important to early development in *Drosopihla* embryos, anterior and posteriorly deposited maternal factors, such as *Bicoid* and *Nanos* respectively. The initial conditions contain a nonzero mRNA concentration in either the most anterior or posterior nucleus at the initial time point, zero initial mRNA concentrations at all other spatial locations, and zero initial protein concentrations in all nuclei.

**A.**

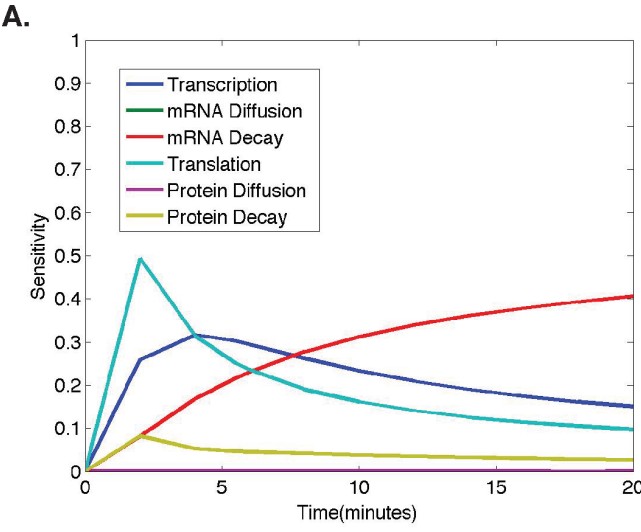

**B.**

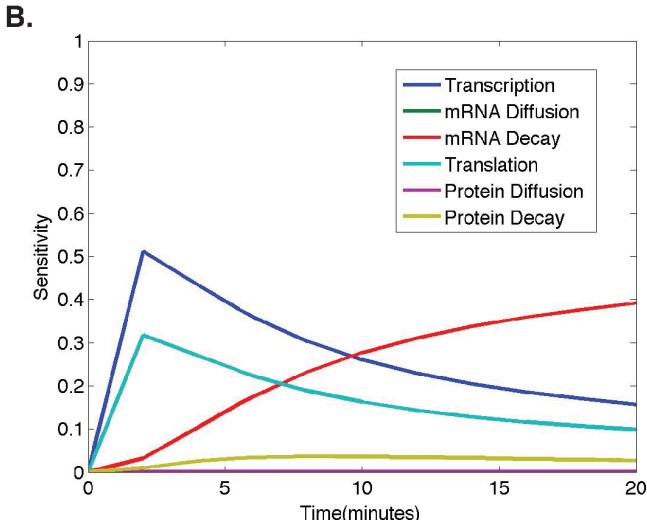

**Figure 4** **Temporal dynamics of parameter sensitivities.** (A) First-order parameter sensitivities at the middle nucleus over time for a ubiquitous gene with initial concentrations of 1.0. (B) First-order parameter sensitivities at the middle nucleus over time for an anterior maternally deposited gene.

## Exploring parameter space

Before a global parameter sensitivity analysis can be performed, one must first define parameter space. This is done by finding a range for each model parameter that results in 'realistic' model predictions. This will result in a six-dimensional hypercube, as required for the Sobol Set sampling method (*Sobol, 1976*). For our model, we determine this range for each parameter by exploring six-dimensional parameter space and recording the parameter value combinations resulting in model predictions of reasonable protein concentrations.

The exploration of parameter space is done iteratively on a parameter-by-parameter basis. First, we define all protein concentration values $\geq 7$ as protein saturation and $\leq 0.01$ as undetectable protein. The method then works in the following way:

1. Initial parameter ranges are set to $\sigma, d, \lambda, \delta \in [0, 1]$, $\tau \in [0.012, 1]$, and $\gamma \in [0.2248, 1]$. Note: Lower bounds were chosen using the values estimated in *Dresch et al. (2013)*.

2. Choose a single parameter. Test each of the boundaries of the parameter range by holding the parameter of interest constant at the lower (or upper) bound and varying all other parameters within their current ranges. Then, run all simulations for $t \in [0, 10]$ with all 5 different initial conditions. If >5% of all simulations result in saturated or undetectable protein concentrations, then modify the boundary of the parameter range by increasing or decreasing it by a number between 0.001 and 0.1.
   Note: Lower bounds on parameter ranges were never allowed to go below 0.0, as negative parameter values would violate the model assumptions.

3. Repeat step 2 for all remaining parameters.

4. When all parameter ranges have been tested, let all 6 parameters vary within their current ranges. If > 5% of all simulations result in saturated or undetectable protein concentrations, then go back to step 2. If not, then stop.

The realistic parameter ranges we obtained and used for our subsequent sensitivity analyses are listed in Table 1.

## Objective function

To calculate model parameter sensitivies, we utilize a previously developed HDMR MATLAB script (*Ziehn & Tomlin, 2009*). Due to the spatial and temporal dynamics of our model, we perform this analysis using a variety of different objective functions.

We focus our analysis on a twenty minute time window in an early blastoderm embryo. Within this time window, we perform our sensitivity analysis at eleven distinct, evenly distributed time points. This removes any bias in the time point chosen and allows us to look at the behavior of parameter sensitivities over time.

One should note that a parameter's first-order sensitivity is calculated by approximating the variance with respect to that parameter divided by the total variance of the objective function. Thus, at $t = 0$, this results in a ratio of $\frac{0}{0}$ since all parameter sets will result in an output equal to the initial condition. Thus, we define the first-order sensitivity of each parameter to be zero when the model output used is from the time point $t = 0$.

Spatially, the model aproximates mRNA and protein concentrations at 52 nuclei across the anterio-posterior axis of the embryo. When performing our sensitivity analysis, we focus on protein concentrations only and consider three different spatial locations: a nucleus in the anterior of the embryo, in the posterior of the embryo, or in the center of the embryo. We also perform sensitivity analyses using the mean protein concentration over all 52 nuclei.

To avoid any inherent bias in our results, we compute parameter sensitivities using each of the five initial conditions and each combination of spatial and temporal protein concentrations. Thus, the results shown are represenatitive of 120 runs of the HDMR algorithm.
## COMPARING MODEL SENSITIVITIES TO EXPERIMENTAL DATA

We begin our analysis by comparing the sensitivities for all six model parameters to previously defined contributions of these processes (*Li, Bickel & Biggin, 2014*). For all inititial conditions, we identify a time point in development such that our model parameter sensitivities are qualitatively similarly to the calculated contribution to protein abundance from *Li, Bickel & Biggin (2014)* (Fig. 2). However, we do observe minimal quantitative differences. When modeling a ubiquitous gene with initial concentrations of one and calculating sensitivities using model output at $t = 4$, we find that the first-order sensitivities differ by a maximum of $\pm 0.09$, with a Pearson correlation coefficient of $> 0.96$ between the first-order sensitivities and the calculated contribution to variance in protein levels from *Li, Bickel & Biggin (2014)*. When modeling an anterior maternally deposited gene and calculating sensitivities using model output at $t = 2$, we find that the first-order sensitivities differ by a maximum of $\pm 0.15$, with a Pearson correlation coefficient of $> 0.89$ between the first-order sensitivities and the calculated contribution to variance in protein levels from *Li, Bickel & Biggin (2014)*. These small observed differences could be due to a number of biological reasons, including noise in the experimental data, species to species variation, or variation in the time point in development in which different genes are expressed. The sensitivities in Fig. 2 are similar to those obtained for all other runs of the model tested (i.e., posterior maternally deposited and ubiquitous genes with other initial concentration values).

To determine whether or not parameter interactions play a significant role in the model's behavior, in addition to computing first-order sensitivites, we also consider second-order and total sensitivities (Fig. 2). Here, note that second-order sensitivities account for a very small portion ($< 12\%$) of the total sensitivity for each parameter. Including second-order sensitivities allows us to account for 99% of total model sensitivity. Therefore, as has been done in past implementations of HDMR, the sum of first- and second-order sensitivities is shown as an approximation to the total sensitivity of each parameter (*Li, Rosenthal & Rabitz, 2001*; *Liang & Guo, 2003*; *Dresch et al., 2010*). However, one should note that first-order sensitivities alone account for over 85% of total sensitivity. This indicates that a great deal of information regarding the contribution of each parameter to the overall behavior of the gene expression system can be gleaned from the first-order sensitivities, which are in strong agreement with the experimental data from *Li, Bickel & Biggin (2014)* (Fig. 2).

## DYNAMIC PARAMETER SENSITIVITIES

Due to the dynamic nature of *Drosophila* development, we also analyze parameter sensitivities at multiple different time points. Figure 3 contains first-order sensitivities corresponding to model simulations with the same initial conditions as those used in the analyses of a ubiquitously expressed and an anteriorly-deposited mRNA in Fig. 2, but calculated using model output at a later time point ($t = 10$ in both cases). One should note that qualitatively, the parameter sensitivities have changed drastically. The sensitivity of $\lambda$, the parameter representing mRNA decay, has increased significantly in both cases (from 0.17 to 0.31

for a ubiquitous gene and 0.03 to 0.28 for an anterior deposited gene) while the sensitivities of both $\sigma$ and $\tau$, parameters representing transcription and translation, have decreased. This stark difference from the sensitivities shown in Fig. 2 suggests that one should further investigate the dynamic nature of the parameter sensitivites with respect to this model.

To better understand how parameter sensitivities are changing over the twenty minute time interval in which we run the model, we choose eleven different evenly distributed time points ($t = 0, 2, 4, 6, 8, 10, 12, 14, 16, 18, 20$). For each set of initial conditions, the HDMR algorithm is implemented using protein concentrations from each of the eleven time points to calculate the parameter sensitivities. The results corresponding to model simulations with the same initial conditions as those used for the analysis in Figs. 2–3 are shown in Fig. 4 for each model parameter as a function of the time point used for the HDMR analysis.

When considering the dynamics of parameter sensitivities, the general trend observed in all results obtained using nonzero initial conditions is that the model is most sensitive to changes in $\sigma$ and $\tau$, parameters representing transcription and translation, at earlier time points, and becomes more sensitive to $\lambda$, the parameter representing mRNA decay, at later time points (Fig. 4).

In Fig. 4A, the model begins with nonzero mRNA and protein concentrations across all nuclei. Thus, $\sigma$ (transcription) is important, but the model is more sensitive to $\tau$ (translation) initially as translation is increasing the protein concentrations at a rate proportional to the nonzero mRNA concentration (Fig. 4). One should also note that the mRNA concentration is increasing at a rate equal to $\sigma$ (transcription) and decreasing at a rate proportional to the mRNA concentration, causing the model to become more sensitive to $\lambda$ (mRNA decay) and $\sigma$ (transcription) at later time points ($t > 5$) (Fig. 4).

In Fig. 4B, we observe a slightly different trend in the parameter sensitivities during early time points due to the fact that initial mRNA concentrations are zero in all nuclei, except the single most anterior nucleus and protein concentrations are zero in all nuclei. Due to the initial mRNA and protein concentrations of zero in the middle nucleus, and the fact that protein can only be synthesized if the mRNA concentration is greater than zero, the model is more sensitive to $\sigma$ (transcription) than $\tau$ (translation) at early times points. One should note that although the model predictions are quite distinct for the different initial conditions, both predictions show mRNA and protein concentrations in the middle nucleus eventually approaching equilibrium values, and the model exhibits similar relative parameter sensitivities at later time points (Fig. 4). Regardless of the qualitative similarities, it is important to note that the dynamic parameter sensitivities are dependent on the initial conditions of the system (Fig. 4).

In both cases shown here, the higher model sensitivity with respect to $\sigma$ (transcription) than $\tau$ (translation) at mid to late time points is in agreement with the conclusion of Li et al. that transcription explains the largest percentage of variance in true protein levels (*Li, Bickel & Biggin, 2014*). However, the large contribution from mRNA decay at late time points was not found in the Li et al. study, as they were unable to consider any dynamic protein levels.

## DISCUSSION

To develop a deeper understanding of the mechanisms involved in regulating the levels of protein concentrations during early development in metazoans, one must consider not only the experimental data that has been carefully collected in the lab, but also the power of mathematical models and the biological interpretation of the parameter values that they use (*Dresch et al., 2010*; *Ay & Arnosti, 2011*). A few very important aspects of modeling that one must consider are how well the model can simulate the reality of the overall system, how well the model agrees with the parts of the system that are already defined biologically, and whether the model parameters can be interpreted in terms of the real biological phenomena that they are assumed to represent. The last of these points is one of the most important, yet remains absent from many modeling studies. In this study, we have used parameter sensitivity analysis to try to unravel the importance of parameter values in a reaction–diffusion model and in doing so to better understand the power of this modeling approach as well as the relative contribution of transcription and translation in regulating protein abundance.

The relative contribution of transcription, translation, and decay rates in overall protein abundance can be aproximated using experimental data (*Schwanhausser et al., 2011*; *Li, Bickel & Biggin, 2014*); however, we ask the question of whether these relative contributions match those found using a mathematical model of gene regulation. We find that the relative parameter sensitivities are in close agreement with the contributions found experimentally (*Li, Bickel & Biggin, 2014*) for various initial conditions at particular time points (Fig. 1). This leads us to believe that even this simple reaction–diffusion model is capturing the correct overall dynamics for a gene with spatially uniform transcriptional activity.

A number of recent studies have directly addressed the dynamics of gene expression in the model system of the developing *Drosophila* embryo through quantitative imaging approaches. In *Drosophila*, the transcriptional activation of the *hunchback* gene by the BICOID protein in the anterior half of the pre-cellular embryo is itself relatively static from nuclear cycle 10 until mid-nuclear cycle 14 (approximately 70 min) in regions where there is a high concentration of BICOID (*Garcia et al., 2013*; *Lucas et al., 2013*). However, at the posterior limit of *hunchback* expression, where the BICOID concentration is lower, there is stochastic on/off transcription, suggesting a threshold level of BICOID is required to initiate transcriptional activation. In contrast, the regulation of *even-skipped* transcription from the stripe 2 enhancer is known to be very dynamic (*Small, Blair & Levine, 1992*). Although initiated in a broad expression domain in nuclear cycle 11 and 12, transcription becomes increasingly refined in nuclear cycle 13 and 14 to produce a single mature stripe only 2 or 3 nuclei wide. Live imaging studies recently confirmed the dynamic nature of expression directed by the stripe 2 enhancer and demonstrated that the mature stripe is also surprisingly transient (as transcription is lost within 30 min, by the end of nuclear cycle 14), with individual nuclei exhibiting discontinuous bursts of transcription (*Bothma et al., 2014*). These results emphasize the need to carefully consider the importance of the dynamic spatial and temporal characteristics of gene expression in the networks that regulate embryonic patterning.

Due to the dynamic nature of the *Drosophila* embryo system, and corresponding mathematical model, we investigated the dynamic nature of parameter sensitivities. These results lead to a very important modeling conclusion and a further biological question. First, one should be cautious when computing parameter sensitivites for a model that is dynamic and a system that is not necessarily in equilibrium, as many protein concentrations are extremely dynamic during early development in an organism. Looking at parameter sensitivities at a single time point can lead to a conclusion that holds for that single time point alone, and the overall behavior of the model may be lost in interpreting these sensitivities individually. Here, we have illustrated the importance of conducting a thorough sensitivity analysis across multiple time points. Second, one must remember the underlying biological question: Which is more important, transcription or translation? In light of the result that relative parameter sensitivites are dynamic, one should reconsider whether this question has a single answer. An interesting question that this study has raised is whether there is a trade-off throughout the early development of the organism. With more experimental data, taken over the course of development, one could ask the detailed question in future studies: Is transcription always contributing more to the variance in protein levels or are there certain points in development where the relative contributions shift?

### Funding

The research in this paper was supported by funding to R.A.D. from the National Institutes of Health (GM090167) and the National Science Foundation (IOS-0845103). G.D.M. was funded as a Fellow from a National Science Foundation Interdisciplinary Training Award to Hampshire College (DBI1129152). The funders had no role in study design, data collection and analysis, decision to publish, or preparation of the manuscript.

### Grant Disclosures

The following grant information was disclosed by the authors:
National Institutes of Health: GM090167.
National Science Foundation: IOS-0845103.
National Science Foundation Interdisciplinary Training Award to Hampshire College: DBI1129152.

### Competing Interests

The authors declare that there are no competing interests.

### Author Contributions

- Gregory D. McCarthy conceived and designed the experiments, performed the experiments, analyzed the data, wrote the paper, prepared figures and/or tables, reviewed drafts of the paper.

- Robert A. Drewell conceived and designed the experiments, analyzed the data, contributed reagents/materials/analysis tools, wrote the paper, reviewed drafts of the paper.
- Jacqueline M. Dresch conceived and designed the experiments, analyzed the data, wrote the paper, prepared figures and/or tables, reviewed drafts of the paper.

## Supplemental Information

Supplemental information for this article can be found online at http://dx.doi.org/10.7717/peerj.1022#supplemental-information.

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
