# Peer review of "Global sensitivity analysis of a dynamic model for gene expression in Drosophila embryos"

_PeerJ, doi:10.7717/peerj.1022_

## Round 0.1 · original submission · Major Revisions

Both reviewers propose some positive comments and also raised some questions. The authors are encouraged to address the reviewers concern and submit a revised version along with a point to point response.

·

Basic reporting

By presenting a dynamic mathematical model that incorporates terms representing
several biological processes in an early Drosophila embryo, the authors indicate that transcription and translation may be the key parameters to determine protein abundance. This is an interesting explanation for understanding the complex mechanics behind the protein expressions. The model and the validation results seem effective to me. I suggest to accept this paper for publishing in Peer J if the authors would make some minor revisions as follows.

1. Add more references and corresponding introductions on the dynamic models under the effect of noises or perturbations.

2. Review the literatures on dynamical network biomarkers which present a model-free way to describe the critical shift phenomenon in biological processes.

Experimental design

The experimental design is clear. I wonder if the authors could provide more background and details on their modelling process and parameter selection.

Validity of the findings

The validation seems sufficient to me. I suggest the authors present their numerical simulation source code on the supplementary information so that readers could use their method in the study of parameter sensitivity.

Additional comments

In this work, the authors indicate that transcription and translation may be the key parameters to determine protein abundance by presenting a dynamic mathematical model that incorporates terms representing several biological processes in an early Drosophila embryo. The model and the validation results seem effective to me. I suggest to accept this paper for publishing in Peer J if the authors would make some minor revisions listed above.

Reviewer 2 ·

Basic reporting

No Comments

Experimental design

1.The several key processes are not well described. For example,
(1) how to obtain the parameter ranges in Table 1, the authors should give the details in Section “Exploring the parameter space”.

(2) In using the HDMR algorithm to make the sensitivity analysis, what are the first-order and second –order sensitivities? Related to equation (2), what is fi, fij, etc. How to compute?

2. Please check the last equation in Page 5, I think it is wrong.

Validity of the findings

3. The author said that they performed sensitivity analyses from three different spatial locations, However, the author only presented one of them. I am not clear if the conclusion under the remaining condition combination is consistent with the paper.

---

## Round 0.2 · accepted · Accept

I am glad to see the improvement of the manuscript via incorporating the reviewers' comments. I am happy to Accept the manusrcript.